# Acceleration of Global Optimization Algorithm by Detecting Local Extrema Based on Machine Learning

**DOI:** 10.3390/e23101272

**Published:** 2021-09-28

**Authors:** Konstantin Barkalov, Ilya Lebedev, Evgeny Kozinov

**Affiliations:** Department of Mathematical Software and Supercomputing Technologies, Lobachevsky University, 603950 Nizhny Novgorod, Russia; ilya.lebedev@itmm.unn.ru (I.L.); evgeny.kozinov@itmm.unn.ru (E.K.)

**Keywords:** global optimization, local optimization, multiextremal problems, numerical methods, approximation, decision trees

## Abstract

This paper features the study of global optimization problems and numerical methods of their solution. Such problems are computationally expensive since the objective function can be multi-extremal, nondifferentiable, and, as a rule, given in the form of a “black box”. This study used a deterministic algorithm for finding the global extremum. This algorithm is based neither on the concept of multistart, nor nature-inspired algorithms. The article provides computational rules of the one-dimensional algorithm and the nested optimization scheme which could be applied for solving multidimensional problems. Please note that the solution complexity of global optimization problems essentially depends on the presence of multiple local extrema. In this paper, we apply machine learning methods to identify regions of attraction of local minima. The use of local optimization algorithms in the selected regions can significantly accelerate the convergence of global search as it could reduce the number of search trials in the vicinity of local minima. The results of computational experiments carried out on several hundred global optimization problems of different dimensionalities presented in the paper confirm the effect of accelerated convergence (in terms of the number of search trials required to solve a problem with a given accuracy).

## 1. Introduction

The successful application of machine learning (ML) methods to solve a wide range of problems leads to the emergence of new ways to apply ML for many tasks. Methods of machine learning were shown to be particularly effective for identifying the principal properties of the phenomena (for example, physical, economic, or social), which are stochastic by nature or contain some hidden parameters [1,2]. ML is also successfully used to solve complex problems of computational mathematics, for example, for simulation of dynamical systems [3], solution of ordinary, partial, or stochastic differential equations [4,5,6].

In particular, ML could be applied for solving such a complex problem of computational mathematics as global optimization. The solution to this class of problems, as a rule, cannot be found analytically and, therefore, one needs to construct numerical methods to solve it.

The numerical solution of optimization problems is fraught with significant difficulties. In many ways, they are related to the dimensionality and type of the objective function. Consequently, the most difficult problems are those in which the objective function is multi-extremal, nondifferentiable, and, moreover, given in the form of a “black box” (i.e., in the form of some computational procedure, the input of which is an argument, and the output is the corresponding value of the function). These complex problems are the main focus of this article.

There are several approaches to the construction of numerical methods for solving global optimization problems. Some algorithms are based on the idea of a multistart: launching a local search either from different starting points or with varying parameters. Local optimization methods have a high convergence rate. At the same time, one of the main problems in multistart schemes is the choice of starting points that would correspond to the regions of attraction of various local solutions. Machine learning methods can be successfully applied to solve this problem. For example, in [7] methods of cluster analysis were used to select promising starting points. In [8], the area for starting the local method was allocated based on the classification of starting points using a support vector machine.

Machine learning methods are actively used in combination with bayesian optimization algorithms based on the probabilistic surrogate models of the objective function. The detailed overview of this trend in the development of global optimization methods is presented in [9,10].

Another popular class of methods for solving global optimization problems is metaheuristic algorithms. Many of them are based on imitation of the processes occurring in living nature. The parameters of such algorithms could also be tuned using ML. For example, ref. [11] provides an overview of machine learning applications in evolutionary algorithms.

Please note that the algorithms of the latter class do not provide guaranteed convergence to the solution of the problem and are inferior to deterministic algorithms in terms of the quality of solution [12,13] (e.g., measured by the number of correctly solved problems from a particular set). Therefore, deterministic methods seem to be potentially more effective.

This paper aims to further develop the efficient deterministic global optimization method known as the information-statistical global search algorithm [14]. The book referenced here contains the results of theoretical studies of the method that are of direct importance for its practical implementation. In particular, it discusses in detail the issues of convergence, the choice of parameters, and the conditions for stopping the algorithm, etc. Please note that the global search algorithm was originally designed for solving unconstrained optimization problems. Later, it was generalized to solve problems with non-convex constraints [15] and multicriteria optimization problems [16]. At the same time, scholars proposed various parallel versions of these algorithms, which can be used on modern supercomputers [17,18,19].

Several strategies have been proposed to speedup the global search algorithm (in terms of the number of iterations required to solve the problem with a given accuracy). In this paper, we propose a new approach to acceleration based on identifying areas of attraction of local minima using machine learning methods. The identification of regions of attraction and the launch of local search in these regions can significantly reduce the number of trials required for the method to achieve global convergence. Experiments carried out on a series of several hundred test problems confirm this statement.

## 2. Problem Statement

In this paper, we will consider global optimization problems of the form
(1)$$\begin{array}{cc} & \varphi \left(y^{*}\right)=\min\left\{\varphi (y):y\in D\right\},\\ & D=\left\{y\in R^{N}:a_{i}\leq y_{i}\leq b_{i},1\leq i\leq N\right\}. \end{array}$$
Problem (Equation 1) is considered under the assumption that the objective function is multi-extremal, is given in a form of a “black box”, and the calculation of its values is associated with solving the problem of numerical simulation, which makes the solution a labor-intensive operation.

A typical situation for many applied problems is when a limited change in the vector of parameters *y* causes a limited change in the values of φ(y). The mathematical model describing this premise is based on the assumption that the Lipschitz condition is satisfied
$$\left|\varphi \left(y^{\prime }\right)-\varphi \left(y^{\prime \prime }\right)\right|\leq L∥y^{\prime }-y^{\prime \prime }∥,\;y^{\prime },y^{\prime \prime }\in D,\;0<L<\infty .$$
This assumption is typical for many approaches to the development of optimization algorithms [20,21,22,23,24]. At the same time, many known approaches are based on various methods of dividing the search domain into a system of subdomains and then choosing the most promising subdomain for placing the next trial (calculating the value of the objective function) [25,26,27,28,29,30]. An important property of global optimization problems is the fact that, in contrast to the problems of finding a local extremum, the global minimum is an integral characteristic of the problem being solved. Making sure that the point $y^{*}\in D$ is a solution to the problem requires going beyond its neighborhood to the investigation of the entire search domain. As a result, when minimizing substantially multi-extremal functions, the numerical method must construct a coverage of the search domain. The number of nodes of this coverage increases exponentially with increasing dimensionality. This feature determines the high complexity of solving multiextremal optimization problems making dimensionality a critical factor that affects the complexity of their solving.

The dimensionality in multi-extremal optimization leads to many issues, so, scholars use a wide variety of approaches to reducing it. For example, simplicial or diagonal partition of the search domain allows using methods for solving one-dimensional problems to solve the original multidimensional problem (see, for example, refs. [31,32]). Another well-known approach to dimensionality reduction is using the Peano space-filling curves to map the multidimensional domain onto a one-dimensional interval [14,33].

In this work, we will use another method based on the nested optimization scheme [34,35,36,37] and its generalization [38,39]. The nested optimization scheme, on the one hand, does not worsen the properties of the objective function (unlike reduction using Peano curves), and, on the other hand, does not require the use of complex data structures to support simplex or diagonal partitions of the feasible region. At the same time, the nested optimization scheme makes it possible to reduce the original multidimensional optimization problem to a family of recursively connected one-dimensional optimization subproblems, which can be solved by a wide range of one-dimensional global optimization algorithms.

## 3. Methods

### 3.1. Core Global Search Algorithm

As a standard, let us consider a one-dimensional multiextremal optimization problem:(2)$$\varphi^{*}=\varphi \left(x^{*}\right)=\min\left\{\varphi \left(x\right):x\in \left[a,b\right],a<b,a,b\in R\right\}$$
with an objective function satisfying the Lipschitz condition.

Here is the description of global search algorithm (GSA) for solving the basic problem in accordance with [14]. In the course of its work, GSA generates a sequence of points $x^{i}$, at which the values of the objective function $z^{i}=\varphi \left(x^{i}\right)$ are calculated. We will refer to the process of calculating the value of the objective function as trial.

In accordance with the algorithm, the first two trials are carried out at the boundary points of the segment [a,b], i.e., $x^{0}=a,\;x^{1}=b$. At these points, the values of the objective function $z^{0}=\varphi \left(x^{0}\right),\;z^{1}=\varphi \left(x^{1}\right)$ are calculated and the counter value is set to k=1. The point of the next trial $x^{k+1},k\geq 1,$ is selected in accordance with the following procedure.

Step 1. Renumber (starting at 0) the points $x^{i},\;0\leq i\leq k$, of the trials conducted in ascending order of the coordinate, i.e.
(3)$$a=x_{0}<x_{1}<\ldots <x_{k}=b.$$
Associate the values of the objective function $z_{i}=\varphi \left(x_{i}\right),\;0\leq i\leq k,$ to the points $x_{i},\;0\leq i\leq k$, at which these values were calculated.

Step 2. Calculate the maximum absolute value of the relative first difference
(4)$$\mu =\max_{1\leq i\leq k}\frac{\left|z_{i}-z_{i-1}\right|}{\Delta_{i}},$$
where $\Delta_{i}=x_{i}-x_{i-1}$. If the value calculated in accordance with (Equation 4) is equal to zero, then take μ=1.

Step 3. For all the intervals $(x_{i-1},x_{i}),1\leq i\leq k$, calculate the value
(5)$$R\left(i\right)=r\mu \Delta_{i}+\frac{{\left(z_{i}-z_{i-1}\right)}^{2}}{r\mu \Delta_{i}}-2\left(z_{i}+z_{i-1}\right),$$
refered to as the *characteristics* of the interval; value r>1 is the parameter of the algorithm.

Step 4. Find the interval $\left(x_{t-1},x_{t}\right)$ with the maximum characteristic
(6)$$R\left(t\right)=\max_{1\leq i\leq k}R\left(i\right).$$
If the maximum characteristic corresponds to several intervals, then choose the minimum number that satisfies (Equation 6) as *t*.

Step 5. Carry out a new trial at the point
(7)$$x^{k+1}=\frac{1}{2}\left(x_{t-1}+x_{t}\right)-\frac{z_{t}-z_{t-1}}{2r\mu }.$$

The algorithm stops when the condition $\Delta_{t}<\epsilon$ is satisfied; here *t* is from (Equation 6), and ϵ>0 is a given accuracy. The values
$$z_{k}^{*}=\min_{0\leq i\leq k}\varphi \left(x^{i}\right),\;x_{k}^{*}=\arg\min_{0\leq i\leq k}\varphi \left(x^{i}\right)$$
are selected to estimate the solution.

The theoretical conditions that determine the convergence of the algorithm are presented in [14]. The work of the algorithm during the minimization of a specific multiextremal function, which is specified in accordance with formula (Equation 17), is shown in Figure 1. The algorithm was launched with the parameter r=2.2 from (Equation 5) and value $\epsilon =10^{-3}$ in the method stopping condition. Figure 1 shows the objective function graph and points of 71 search trials which GSA needed to solve the problem to the specified accuracy. This highlights the problem of all methods of global optimization—the concentration of trial points in the vicinity of local minima of the problem, which are not a global solution.

### 3.2. Machine Learning Regression as a Tool for Identifying Attraction Regions of Local Extrema

The functions considered in this study belong to the class of Lipschitzian functions. Therefore, classical regression methods (for example, polynomial regression, where a function is approximated by a polynomial of a given degree) will not properly match the behavior of the function. A more powerful tool for this task is regression splines. When constructing a regression spline, the domain is divided into *K* non-overlapping subdomains. In each of such domains, the function is approximated by a polynomial. Dividing the interval into a sufficient number of subdomains allows one to very accurately approximate the original function.

Regression can also be constructed using such a powerful tool as artificial neural networks. Different types of networks can be used to build the regression, for example, multilayer perceptron, radial basis function network, etc. However, in both of these cases, the model itself (spline or neural network) becomes rather complex for the analysis required to solve the given problem (identifying areas of attraction of local extrema).

Therefore, within the framework of the study, we chose a regression model based on decision trees to analyze the local behavior of a function. For example, if the objective function is properly approximated by polynomials, then the polynomial regression, of course, will appropriately convey the properties of the function. However, if there is a more complex relationship between them, then the decision tree can surpass the classical variants of regression in terms of the quality of the approximation. At the same time, the regression based on the decision tree makes it possible to easily identify the areas of attraction of local extrema with sufficient accuracy.

Building a regression using a decision tree consists of two main steps:Search domain *D* is divided into *J* non-overlapping subdomains $D_{1},D_{2},\ldots ,D_{J}$, provided that $D={⋃}_{j=1}^{J}D_{j}$.Any value falling into the subdomain $D_{j}$, i.e., $x\in D_{j}$, is matched to the average value $c_{j}$ based on the training trials that fall into this subdomain.

In fact, decision trees build a model of a function of the form
(8)$$f\left(x\right)=c_{j},\;x\in D_{j}.$$

Generally speaking, a decision tree is a binary tree, the leave nodes of which contain the values of the function, and the other nodes contain the transition conditions. In our case, when applying a decision tree to construct a regression, each node corresponds to the results of several trials and the value $c_{j}$, which is calculated as the mean square of the trial points assigned to this node. The final piecewise constant approximation is constructed using the values $c_{j}$, located in the leave nodes of the tree.

The tree is built recursively, starting from the root node. A decision rule is applied at each node: all data are divided into two groups (according to a partitioning rule) and sent to the left and right child nodes. The procedure then recursively separates the left and right nodes. Recursion stops at a node in one of the following cases:The number of trial points assigned to the node becomes less than the specified threshold value (we used 1).The sum of the squared deviations of the function values from the value $c_{j}$, assigned to this node becomes less than the set accuracy (we used $10^{-3}$).

For more information on the algorithm for constructing regression using decision trees, see, for example [40].

Regression built using decision trees, is, on the one hand, rather simple, and on the other hand, it adequately reflects the properties of the function under investigation (the presence or absence of local minima). The selection of regions of attraction of local minima using model (Equation 8) can be organized as follows.

Let $x^{k+1}$ be the point of the current trial. For a given point, an index *j* is sought such that $x^{k+1}\in D_{j}$. Next, the $c_{j}$ values corresponding to neighboring subdomains are compared. If one of the conditions is met
(9)$$\begin{array}{c} c_{j}\leq c_{j+1}\leq c_{j+2}\leq c_{j+3}\leq c_{j+4},\;j=1;\\ c_{j-1}\geq c_{j}\leq c_{j+1}\leq c_{j+2}\leq c_{j+3},\;j=2;\\ c_{j-2}\geq c_{j-1}\geq c_{j}\leq c_{j+1}\leq c_{j+2},\;3\leq j\leq J-2;\\ c_{j-3}\geq c_{j-2}\geq c_{j-1}\geq c_{j}\leq c_{j+1},\;j=J-1;\\ c_{j-4}\geq c_{j-3}\geq c_{j-2}\geq c_{j-1}\geq c_{j},\;j=J; \end{array}$$
then the subdomain $D_{j}$ is considered the area of attraction of the local minimum. Here, one can start a local search, and subsequently exclude this subdomain from the global search. Any of the zero-order local methods can be used as a local algorithm, for example, golden section search or parabolic interpolation [41].

To modify the global search algorithm from Section 3.1 to exclude the regions of attraction of local minima, we associate each trial point $x^{i}$ obtained during the operation of the algorithm with an additional attribute $q^{i}\in \left\{0,1,2\right\}$, which will characterize the properties of this point. The value $q^{i}=0$ is assigned by default and indicates that point $x^{i}$ is obtained as a result of rule (Equation 7) of the global search algorithm. The value $q^{i}=1$ is assigned to the points obtained as a result of the work of the local method, while the value $q^{i}=2$ corresponds to the point of the local minimum found as a result of the work of the local search.

Using the additional attribute $q^{i},\;0\leq i\leq k,$ makes it possible to distinguish between points obtained during a global ($q^{i}=0$) and a local ($q^{i}=1,2$) search. The intervals, the boundary points of which have the value $q^{i}=1$ or $q^{i}=2$, will be skipped in the global search, i.e., they will not be tried further. While the selected value $q^{i}=2$, which corresponds to the local minimum will allow using these points for checking the stopping criterion. When the rules of the global search signal that a new trial point needs be placed in a given ϵ-neighborhood of one of the found local minima, this will correspond to the end of the search.

Let us now describe a modified global search algorithm that uses decision trees to isolate and exclude areas of attraction of local minima; we will further refer to this algorithm as GSA-DT. Recall that the superscript corresponds to the number of the iteration at which the trial was carried out at this point, and the subscript corresponds to the number of the point in the row (Equation 3).

Steps 1–5 of the GSA–DT algorithm are the same as steps 1–5 of GSA.

Step 6. If $q_{t-1}\in \left\{1,2\right\}$ or $q_{t}\in \left\{1,2\right\}$, then go to check the stopping criterion. Otherwise, go to Step 7.

Step 7. Construct a decision tree based on the results of the trials performed, and obtain the corresponding piecewise constant approximation, which assigns the value $c_{1},c_{2},\ldots ,c_{J}$ to each subdomain $D_{1},D_{2},\ldots ,D_{J}$.

Step 8. For the point $x^{k+1}$ of the current trial, find a number *j* such that $x^{k+1}\in D_{j}$ and check whether the condition (Equation 9) is satisfied. If condition (Equation 9) is satisfied, start a local search in the domain $D_{j}$ from the point $x^{k+1}$. The results of all trials performed during the local search at the points $x^{k+2},\ldots ,x^{k+k_{local}}$ are stored in the information base of the algorithm and are used at subsequent iterations. All these points receive the attribute $q^{i}=1$, $i=k+2,\ldots ,k+k_{local}$. The attribute equal to 2 is assigned to the point corresponding to the found local minimum.

The stopping criterion of the modified algorithm will look as follows. The algorithm stops when one of the following conditions is met:$|x_{t}-x_{t-1}|<\epsilon$;$|x^{k+1}-x_{t-1}|<\epsilon$ and $q_{t-1}=2$;$|x^{k+1}-x_{t}|<\epsilon$ and $q_{t}=2$;
where *t* is from (Equation 6), and ϵ>0 is a given accuracy.

Please note that the specified stopping criterion is checked after Step 6, i.e., after the calculation according to the rules of the global search for the point of the next trial $x^{k+1}$, but before the trial itself is carried out in it. The idea behind this criterion is as follows. The search stops when the interval $\left(x_{t-1},x_{t}\right)$, in which the point $x^{k+1}$ falls, becomes sufficiently small, or when the point $x^{k+1}$ falls into a small neighborhood of one of the found local minima. Such minima in this case will be simultaneously global.

For example, consider the work of the GSA-DT algorithm with minimization of the same multi-extremal function, which is presented in Figure 1. The same parameters were used at the start of the algorithm: the parameter r=2.2 from (Equation 5) and value $\epsilon =10^{-3}$ in the stopping criterion of the method. Figure 2 illustrates the operation of the GSA-DT algorithm. In addition to the graph of the objective function, it shows a piecewise constant approximation of the form (Equation 8) built at the final stage of the search. Black points on the graph correspond to the global search phase, green points correspond to the work of the local method. In total, the GSA-DT method required 49 trials to solve the problem; there was no accumulation of trial points in the vicinity of local minima.

### 3.3. Adaptive Dimension Reduction Scheme

The recursive nested optimization scheme is based on the well-known relation [35]
(10)$$\min_{y\in D}\varphi \left(y\right)=\min_{y_{1}\in \left[a_{1},b_{1}\right]}\min_{y_{2}\in \left[a_{2},b_{2}\right]}\ldots \min_{y_{N}\in \left[a_{N},b_{N}\right]}\varphi \left(y\right),$$
which allows reducing the solution of the original multidimensional problem (Equation 1) to the solution of a family of recursively connected one-dimensional subproblems.

For a formal description of the nested optimization scheme, we introduce a family of functions defined in accordance with the relations
(11)$$\varphi_{N}\left(y_{1},\ldots ,y_{N}\right)☰\varphi \left(y_{1},\ldots ,y_{N}\right),$$
(12)$$\varphi_{i}\left(y_{1},\ldots ,y_{i}\right)=\min_{y_{i+1}\in \left[a_{i+1},b_{i+1}\right]}\varphi_{i+1}\left(y_{1},\ldots ,y_{i},y_{i+1}\right),1\leq i\leq N-1.$$

Then, in accordance with (Equation 10), solving the multidimensional problem (Equation 1) is reduced to solving a one-dimensional problem
(13)$$\varphi^{*}=\min_{y_{1}\in \left[a_{1},b_{1}\right]}\varphi_{1}\left(y_{1}\right).$$
However, each calculation of the value of the function $\varphi_{1}$ at some fixed point $y_{1}$ presupposes the solution of the one-dimensional optimization problem of the second level
(14)$$\varphi_{1}\left(y_{1}\right)=\min_{y_{2}\in \left[a_{2},b_{2}\right]}\varphi_{2}\left(y_{1},y_{2}\right).$$
Calculation of the values of the function $\varphi_{2}$ in turn, requires one-dimensional minimization of the function $\varphi_{3}$ all the way to the solution of the problem
(15)$$\varphi_{N-1}\left(y_{1},\ldots ,y_{N-1}\right)=\min_{y_{N}\in \left[a_{N},b_{N}\right]}\varphi_{N}\left(y_{1},\ldots ,y_{N}\right)$$
at the last level of recursion.

The solution of the set of subproblems arising in the nested optimization scheme (Equation 12) can be organized in different ways. The obvious way (described in detail in [35,37]) is based on solving subproblems in accordance with the recursive order of their generation. However, a significant part of the information about the objective function is lost here.

Another approach is an adaptive scheme, in which all subtasks are solved simultaneously, which allows taking into account much more information about a multidimensional problem, thereby speeding up the process of its solution. This approach was theoretically substantiated and tested in [38,39,42].

Please note that within the framework of the original nested optimization scheme, the generated subproblems are solved only sequentially; the resulting hierarchical scheme for generating and solving subproblems has the form of a tree. The construction of this tree occurs dynamically in the process of solving the original problem (Equation 1). In this case, the calculation of one value of the function $\varphi_{i}\left(y_{1},y_{2},\ldots ,y_{i}\right)$ at the *i*-th level requires a complete solution of all problems of one of the subtrees of level i+1.

The adaptive nested optimization scheme of dimensionality reduction changes the order of solving subproblems: they will be solved not one by one (in accordance with their hierarchy in the problem tree), but simultaneously, i.e., there will be a set of subtasks in the process of solution. Within the adaptive scheme:to calculate the value of *i*-th level function from (Equation 12) a new i+1 level problem is generated, in which only one trial is carried out, after which the new generated problem is included in the set of already existing problems to be solved;iteration of the global search consists of choosing one (most promising) problem from the set of available problems, in which one trial is carried out; the new trial point is determined according to the basic global search algorithm from Section 3.1 or a modified algorithm from Section 3.2;the minimum values of functions from (Equation 12) are their current estimates obtained based on accumulated search information.

## 4. Experimental Results

Numerical experiments were performed on the Lobachevsky supercomputer of the University of Nizhny Novgorod (operating system CentOS 7.2, management system SLURM). One supercomputer node has two Intel Sandy Bridge E5-2660 2.2 GHz processors, 64 Gb RAM. The CPU is 8-core (i.e., a total of 16 CPU cores are available on the node). All the algorithms were implemented in C++; GCC 5.5.0 was used for compilation on the supercomputer.

The traditional approach to assessing the effectiveness of global optimization methods is based on using these methods to find the numerical solution of a series of problems. In this case, the assumption is that a certain algorithm is used to generate the next problem to be solved. Typical examples of such test function classes are Shekel and Hill functions. The first of them (denoted $F_{SH}$) is based on the formula
(16)$$\varphi \left(x\right)=-\sum_{j=1}^{10}\frac{1}{\left(K_{j}{\left(x-A_{j}\right)}^{2}+C_{j}\right)},\;x\in \left[0,10\right],$$
where parameters $1\leq K_{j}\leq 3,\;0<A_{j},\;C_{j}<10,\;$ are independent random variables uniformly distributed in the indicated intervals. The next generator (denoted $F_{HL}$) is determined by the expression
(17)$$\varphi \left(x\right)=\sum_{j=1}^{14}\left(A_{j}\sin\left(2j\pi x\right)+B_{j}\cos\left(2j\pi x\right)\right),\;x\in \left[0,1\right],$$
where the values of the parameters $A_{j},\;B_{j},\;1\leq j\leq 14$, are independently and uniformly distributed in the interval [−1,1].

Let us compare the basic global search algorithm (GSA) and its decision tree-based modification (GSA-DT) with the well-known DIRECT global optimization algorithm [25]. The choice of this particular method for comparison is explained as follows. DIRECT is one of the most well-known and popular deterministic methods for solving global optimization problems with a “black box” objective function. An overview of various modifications of the method, as well as examples of solving problems, is given in [43]. It is known that with a sufficiently large number of search trials, DIRECT is guaranteed to find a global solution to the problem. However, if the stopping by accuracy is used as a criterion, then the method can abruptly stop at one of the local minima, which is confirmed by the experimental results presented in this section.

Regarding the deterministic optimization algorithms implemented in popular Computer Algebra Systems (CAS), their comparison with methods such as DIRECT or GSA will be incorrect for the following reason. Optimization algorithms from CAS are focused on solving problems with an objective function set explicitly, in the form of a formula. Formula definition of the objective function assumes that its derivatives are also known, which makes it possible to use first-order methods with significantly faster convergence than zero-order methods, which include both DIRECT and GSA. However, first-order methods do not guarantee finding the global minimum. For example, the Mathematica system offers methods for solving global optimization problems that are guaranteed to find a global solution to the problem only if the objective function and constraints are linear or convex. Otherwise, the result may sometimes only be a local minimum.

Moreover, methods that require a formulaic specification of the objective function cannot be applied to the solution of a large class of applied problems in which the form of the objective function is not known, and its values are calculated as a result of numerical simulation. The use of heuristic methods for solving problems of this kind (such as Differential Evolution, Simulated Annealing or Random Search) also does not always lead to the solution of the problem. These methods often do find the global minimum, but are not guaranteed to do so. In terms of the number of correctly solved problems, heuristic methods are inferior to deterministic ones [13].

The decision trees in the GSA-DT algorithm were built using the OpenCV 4.5.1 library (class cv::ml::DTrees). A regression function was constructed by a single decision tree. This allowed us to obtain a piecewise constant approximation of the objective function in which one or several trial points corresponded to each leaf node of the tree. The tree was built without any limitations on maximum depth (MaxDepth); the accuracy of tree construction (RegressionAccuracy) was $10^{-3}$; the minimum number of trial points in the tree node (MinSampleCount) was equal to one; all other parameters were set by default. Stopping tree building occurred if all absolute differences between an estimated value in a node and values of train samples in this node are less than accuracy.

The global optimization methods discussed above were compared when solving 100 problems from the $F_{SH}$ and $F_{HL}$ classes. The problem will be considered correctly solved if after stopping the method by accuracy (i.e., when the length of the current search interval becomes less than $\epsilon \cdot \left|b-a\right|$) the current estimate of the optimum $x_{k}^{*}$ lies in the ϵ-neighborhood of the known solution of the problem $x^{*}$, i.e., if the condition $|x^{*}-x_{k}^{*}|\leq \epsilon \cdot \left|b-a\right|$ is satisfied.

Table 1 and Table 2 show the number of search trials that, on average, were required to minimize the Shekel and Hill functions with different search accuracy ϵ. The number of unsolved problems is indicated in parentheses. These and subsequent tables feature the total number of trials that were performed during both global and local searches by GSA-DT. When solving problems with an accuracy $10^{-2}$, $10^{-3}$, $10^{-4}$ the number of local search launches when solving one problem on average was equal to 2, 3, and 4, respectively; and the ratio of the number of trials performed according to the global search rules to the number of trials performed by the local method was (with appropriate accuracy) 7.2, 6.0, 4.5 for problems of the $F_{HL}$ class and 3.2, 2.3, 1.9 for the $F_{SH}$ class problems.

The experimental results show that with a rough solution to the problem, all methods show similar results in terms of the number of trials, while with a high solution accuracy, the GSA-DT algorithm requires two times fewer trials than its prototype. At the same time, GSA-DT outperforms the DIRECT method both in the average number of search trials and in the number of correctly solved problems. In particular, if we use the accuracy $\epsilon =10^{-2}\left|b-a\right|$, then the DIRECT method stops too early and does not find a global solution to many problems. Therefore, in further experiments in which multidimensional problems are solved, we will not use DIRECT for comparison, since when solving multidimensional problems with stopping by accuracy, this method will provide a correct solution to no more than 50% of problems.

**Table 1 entropy-23-01272-t001:** The average number of tests when minimizing Shekel test functions (the number of unsolved problems is indicated in parentheses).

	$\epsilon =10^{-4}$	$\epsilon =10^{-3}$	$\epsilon =10^{-2}$
DIRECT	64(1)	34(6)	20(17)
GSA	106	53	31
GSA-DT	49	43	35

**Table 2 entropy-23-01272-t002:** The average number of tests when minimizing Hill test functions (the number of unsolved problems is indicated in parentheses).

	$\epsilon =10^{-4}$	$\epsilon =10^{-3}$	$\epsilon =10^{-2}$
DIRECT	66(12)	36(31)	20(51)
GSA	130	75	43
GSA-DT	64	59	50

The next series of experiments involved the solution of multidimensional problems. A well-known generator of multi-extremal optimization test problems is GLKS [44]. It can be used to generate test functions with given properties: the number of local extrema, their areas of attraction, the global minimum point, and the value of the objective function at this point, etc. The procedure for generating test functions is based on using polynomials to redefine a convex quadratic function (paraboloid). Test functions are defined by five parameters:dimensionality of the problem *N*;the number of local minima *l*;value of the global minimum $f^{*}$;radius of the area of attraction of the global optimizer $\rho^{*}$;the distance between the global optimizer and the vertex of the paraboloid $d^{*}$.

By changing the specified parameters, one can create test classes with different properties. For example, with a fixed dimensionality of the problem and the number of local minima, a more complex class can be generated by narrowing the region of attraction of the point of the global minimum or by increasing the distance between this point and the vertex of the paraboloid. In the experiments, the values l=10, $f^{*}=-1$, $\rho^{*}=0.2$ and $d^{*}=0.9$ were used.

As an example, consider the operation of the GSA and GSA-DT algorithms when solving one of the two-dimensional problems generated by the GKLS generator. The level lines of the objective function shown in Figure 3 and Figure 4, indicate the presence of ten local extrema. When starting the algorithms, the same parameters were used: r=3.0 from (Equation 5) and $\epsilon =10^{-2}\left|b-a\right|$ in the stopping criterion of the method. Black dots in Figure 3 and Figure 4 show the points of the search trials performed by the methods in the process of solving the problem. In this case, the GSA algorithm required 247 trials, while the GSA-DT algorithm took 138 trials. The red dot in the figures marks the exact solution of the problem, and the yellow dot indicates the best approximation found by the algorithm. Green dots in Figure 4 indicate trials performed as part of a local search. This graph demonstrates that using decision trees to identify areas of attraction of local minima removes the problem of accumulation of test points in the region of local extrema inherent in the original global search algorithm.

We used GKLS to generate 300 test problems of dimensionalities N=2,3,4 (100 problems of each dimensionality). The resulting series of problems were solved using the GSA and GSA-DT algorithms with the parameter r=5.0 from (Equation 5). The specified value of the parameter *r* ensures the solution of 100% of the problems; at lower values of the parameter, some problems were not solved correctly.

Table 3 and Table 4 show the average number of trials required by the GSA and GSA-DT methods to correctly solve all problems with an accuracy of $\epsilon =10^{-2}\left|b-a\right|$ and $\epsilon =2\times 10^{-3}\left|b-a\right|$, respectively.

The data from the tables confirm that the global search algorithm based on the application of machine learning to extract local extremum provides a faster solution to multiextremum problems than the basic global search algorithm. For rough accuracy solutions, the acceleration is about 30%, for high accuracy solutions, the process is accelerated from 2 to 6 times.

**Table 3 entropy-23-01272-t003:** Solving GKLS problems with an accuracy $\epsilon =10^{-2}\left|b-a\right|$.

	N=2	N=3	N=4
GSA	937	12716	206869
GSA-DT	653	9204	156190

**Table 4 entropy-23-01272-t004:** Solving GKLS problems with an accuracy $\epsilon =2\times 10^{-3}\left|b-a\right|$.

	N=2	N=3	N=4
GSA	1489	69764	583903
GSA-DT	831	10776	173155

## 5. Conclusions and Future Work

The article discusses an efficient deterministic method for solving multiextremal optimization problems—the information-statistical global search algorithm. A new way of speeding up the operation of this algorithm was proposed (in terms of the number of trials required to solve the problem with a given accuracy). This method is based on identifying areas of attraction of local minima of the objective function using machine learning methods. The identification of regions of attraction and the launch of local search in these regions can significantly reduce the number of trials required for the method to achieve global convergence. Within the framework of the investigated approach, solving multidimensional problems is reduced to solving a series of information-related one-dimensional subproblems; therefore, the key point is to identify local minima in one-dimensional problems. This is achieved by approximation of the objective function built using decision trees. Computational experiments were carried out on a series of test problems of different dimensions to compare the speed of the original global search algorithm (GSA) and its modification, which uses decision trees to identify local minima of the objective function (GSA-DT). The experimental results show that the use of the GSA-DT algorithm can significantly (up to 6 times) reduce the number of trials required to solve the problem with a given accuracy.

Further research into this issue will focus on using more complex models of the objective function to obtain a more accurate approximation. We plan to use artificial neural networks as such an approximator. This will require the development of new methods for identifying local extrema since function approximation using a neural network is more complicated from this point of view. We also plan to pay attention to the issues of the reliability of the results obtained using machine learning methods. For the solved model problems, the use of machine learning methods shows good results, but the question of whether this effect will persist in more complex problems remains open.

Another direction for further work will be the combination of the proposed approach to the application of local search with traditional methods of accelerating global optimization algorithms. For example, a well-known way to speed up algorithms of this class is to use local adaptive estimates of the Lipschitz constant $L_{i}$ in various subdomains $D_{i}\in D$ within the search domain *D* instead of a single global estimate of the constant *L* for the entire domain *D* ( see [45,46]). This allows the algorithm to adjust to the local behavior of the objective function in different parts of the feasible domain, and, thereby, reduces the number of search trials required to achieve convergence. Using only global information about the behavior of the objective function during its minimization can significantly slow down the convergence of the algorithm to a point of global minimum. Further research on this issue could focus on using machine learning methods to isolate subdomains $D_{i}\in D$ and construct local estimates of the Lipschitz constant $L_{i}$ in the selected subdomains.

## Figures and Tables

**Figure 1 entropy-23-01272-f001:**
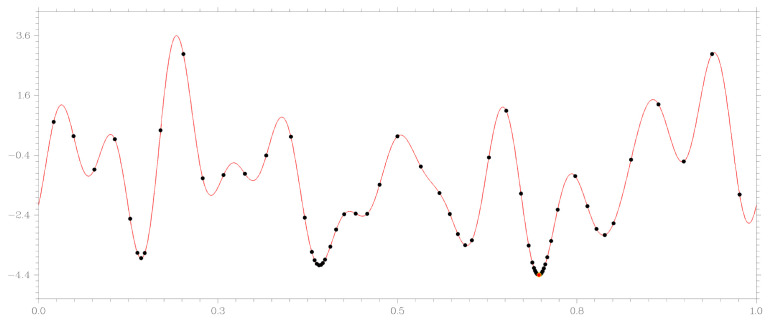
Minimization of a function of the form (Equation 17) using GSA.

**Figure 2 entropy-23-01272-f002:**
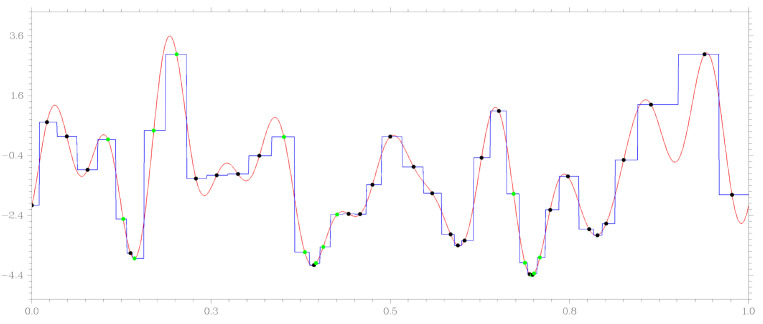
Using GSA-DT to minimize the function (Equation 17).

**Figure 3 entropy-23-01272-f003:**
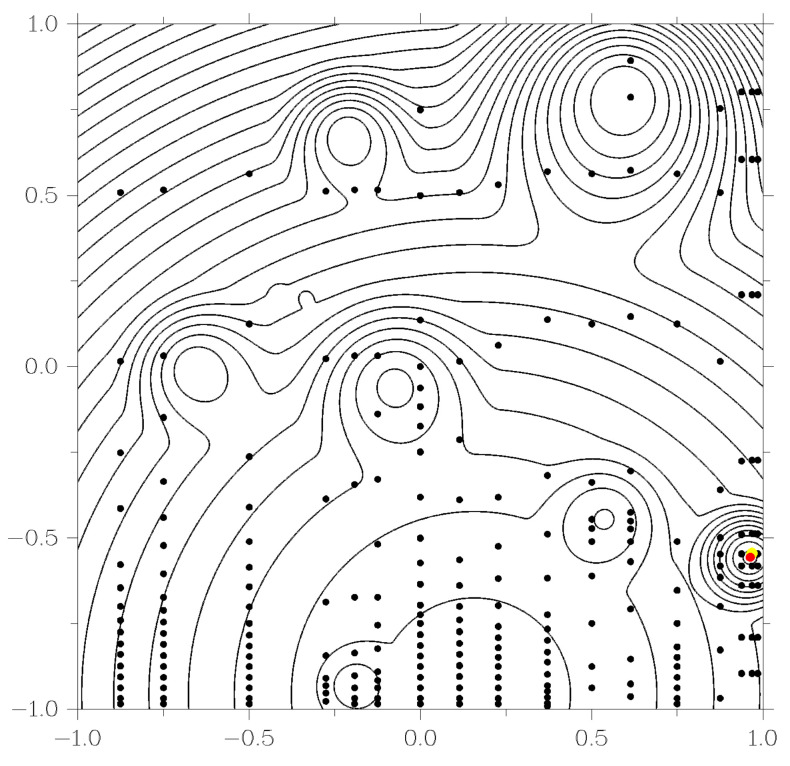
Using GSA algorithm to solve GKLS problems.

**Figure 4 entropy-23-01272-f004:**
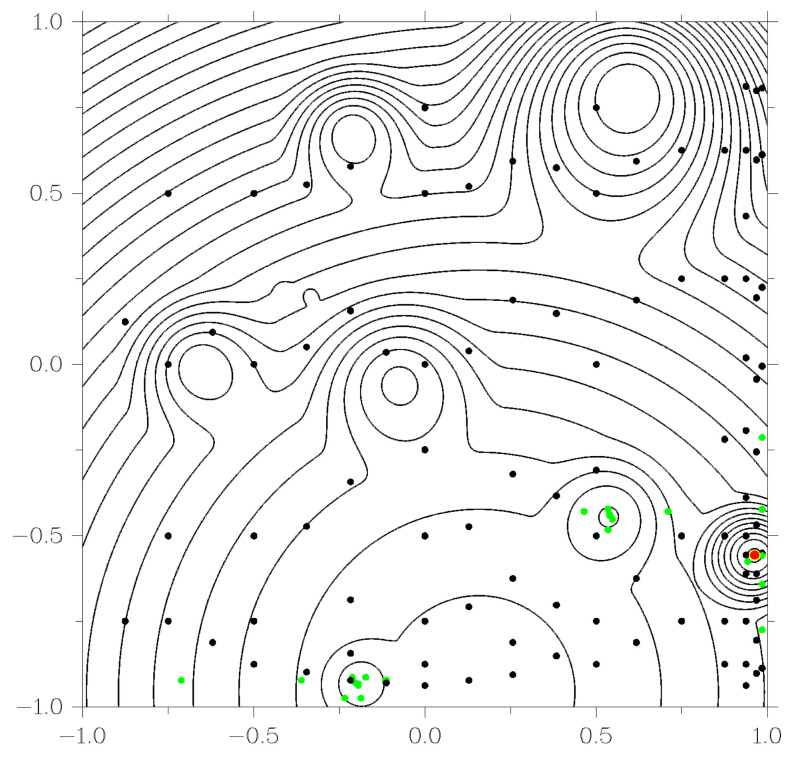
Using GSA-DT algorithm to solve GKLS problems.

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
