# Peer review of "Acceleration of Global Optimization Algorithm by Detecting Local Extrema Based on Machine Learning"

_entropy, 2021, doi:10.3390/e23101272_

Round 1

Reviewer 1 Report

This interesting paper considers an algorithm for searching the global minimum of a black-box-function. The authors suggest using machine learning methods to identify regions of attraction of local minima within the global search algorithm. The use of local optimization algorithms in the selected regions can significantly accelerate the convergence of global search. The results of computational experiments carried out on several hundred global optimization problems of different dimensionalities presented in the paper confirm the effect of accelerated convergence (in terms of the number of search trials required to solve a problem with a given accuracy).

The paper is clearly written and presents novel theoretical and experimental results suitable for publication in this special issue with the following minor modifications.

  • To extend the survey of the machine learning and global optimization methods, the authors can mention at least the following recent publications in the “Introduction”: https://www.springer.com/gp/book/9783030244934, https://www.springer.com/gp/book/9783030647117 , https://www.sciencedirect.com/science/article/pii/S0957417419307699, https://link.springer.com/article/10.1007/s00500-020-05030-3
  • In the “Experimental results” section, please indicate the average ratio of the number of trials performed by the global search algorithm to the number of trials performed by the local method.
  • Indicate also the average number of runs of the local method when solving a series of problems.
  • In Bibliography, initials of some authors are missing, for example, "Sergeyev, Y." should be "Sergeyev, Y.D."; "Mukhametzhanov, M." should be "Mukhametzhanov, M.S."; "Kvasov, D." should be "Kvasov, D.E.", etc. Reference [30] is out of the page width.

Author Response

Answers to Reviewer 1

Reviewer 1. To extend the survey of the machine learning and global optimization methods, the authors can mention at least the following recent publications in the “Introduction”: https://www.springer.com/gp/book/9783030244934, https://www.springer.com/gp/book/9783030647117, https://www.sciencedirect.com/science/article/pii/S0957417419307699, https://link.springer.com/article/10.1007/s00500-020-05030-3

Answer. The survey of the machine learning and global optimization methods is extended, the mentioned references are included in the text (see Introduction).

Reviewer 1. In the “Experimental results” section, please indicate the average ratio of the number of trials performed by the global search algorithm to the number of trials performed by the local method. Indicate also the average number of runs of the local method when solving a series of problems.

Answer. Done, see page 10, paragraph 3.

Reviewer 1. In Bibliography, initials of some authors are missing, for example, "Sergeyev, Y." should be "Sergeyev, Y.D."; "Mukhametzhanov, M." should be "Mukhametzhanov, M.S."; "Kvasov, D." should be "Kvasov, D.E.", etc. Reference [30] is out of the page width.

Answer. Corrected, see the list of references.

Reviewer 2 Report

The paper is dedicated to multidimensional global optimization problems with box constraints, where the objective function is given as a black box. 
A new acceleration method is proposed using a well-known global search algorithm, decision trees and local optimization. Numerical experiments on massive randomized classes of test problems show the advantages of the proposed acceleration technique.  The paper is well-written, well-structured and can be published after a revision following several very minor remarks given below.

  1. I recommend to eliminate the full description of the GSA, since it was well described in many sources. Probably, a short description with the relative references can be made.
  2. How c_j are calculated in (8)? I suggest adding an explicit formula or describing this point better.
  3. Algorithm GSA-DT should be described better and emphasized into a separate environment, defining also better the additional parameters (e.g., the value q). In particular, it is not clear if the first stopping criterion is checked before the step 6 or after? 
  4. For a future research, it can be reasonable to apply the proposed scheme for an algorithm with local adaptive estimates of the Lipschitz constant, which is already considered as an acceleration technique usually. 
  5. Please, state explicitly if the trials obtained by the local optimization are counted in the final tables.

To conclude, the topic of the paper is very interesting from the practical point of view, so I recommend accepting this contribution. 

Author Response

Answers to Reviewer 2

Reviewer 2. I recommend to eliminate the full description of the GSA, since it was well described in many sources. Probably, a short description with the relative references can be made.

Answer. We agree that the description of the global search algorithm is well known. But it seems for us correct to leave it in the text, since it is required for the subsequent description of the algorithm GSA-DT.

Reviewer 2. How  are calculated in (8)? I suggest adding an explicit formula or describing this point better.

Answer. The explanation is given on page 5, last paragraph.

Reviewer 2. Algorithm GSA-DT should be described better and emphasized into a separate environment, defining also better the additional parameters (e.g., the value ). In particular, it is not clear if the first stopping criterion is checked before the step 6 or after?

Answer. Comments about the parameter  and the stopping criterion are included in the text (see page 6, lines 203-210, page 7, lines 235-240)

Reviewer 2. For a future research, it can be reasonable to apply the proposed scheme for an algorithm with local adaptive estimates of the Lipschitz constant, which is already considered as an acceleration technique usually.

Answer. The use of local adaptive estimates of the Lipschitz constant will be one of the directions for future work. The corresponding comment and relevant references are included in the text (see the end of the section Conclusions and Future Work).

Reviewer 2. Please, state explicitly if the trials obtained by the local optimization are counted in the final tables.

Answer. The corresponding comment is included in the text (see page 10, lines 336-338)

Reviewer 3 Report

The paper is interesting. The problem of global optimization arises everywhere in applications, and it is crucial to have a good algorithm to solve it. The GSA (global search algorithm) is one of the algorithms that is used for this problem, and the authors demonstrated how to use machine learning to significantly accelerate it. 

I have only few comments. First, there are many algorithms for global optimization, both deterministic and randomised ones, but the authors use only one of them (DIRECT) for the comparison in the 1-dimensional case. No algorithms except of GSA and improved GSA has been analysed in the multidimensional case. It would be interesting to compare the suggested algorithm with the fastest known algorithms for this problem. In particular, some algorithms for global optimization are implemented in popular Computer Algebra Systems (CAS), like Mathematica. It would be interesting to solve the test problems in popular CAS systems and using the suggested method and compare which is faster. In fact, I think that such a comparison can be done on regular PC (not supercomputer) to save cost for supercomputer time. One just needs to design a bit easier test functions, suitable to optimize on PC.

Another short comment: are there any other attempts in the literature to accelerate GSA? If yes, the authors should review them, compare which acceleration (known ones or the suggested one) works better, and whether they can be combined to produce an even faster algorithm.  

Author Response

Answers to Reviewer 3

Reviewer 3. There are many algorithms for global optimization, both deterministic and randomised ones, but the authors use only one of them (DIRECT) for the comparison in the 1-dimensional case. No algorithms except of GSA and improved GSA has been analyzed in the multidimensional case. It would be interesting to compare the suggested algorithm with the fastest known algorithms for this problem.

Answer. Comment about the comparison of the methods is included in the text (see page 9, lines 292-299)

Reviewer 3. In particular, some algorithms for global optimization are implemented in popular Computer Algebra Systems (CAS), like Mathematica. It would be interesting to solve the test problems in popular CAS systems and using the suggested method and compare which is faster.

Answer. Comment about the methods implemented in popular Computer Algebra Systems (like Mathematica) is included in the text (see page 9, lines 300 - 318)

Reviewer 3. Are there any other attempts in the literature to accelerate GSA? If yes, the authors should review them, compare which acceleration (known ones or the suggested one) works better.

Answer. The comment on another acceleration techniques for GSA is given, see page 13, lines 424-436.